# Recording of Chronic Diseases and Adverse Obstetric Outcomes during Hospitalizations for a Delivery in the National Swiss Hospital Medical Statistics Dataset between 2012 and 2018: An Observational Cross-Sectional Study

**DOI:** 10.3390/ijerph19137922

**Published:** 2022-06-28

**Authors:** Carole A. Marxer, Marlene Rauch, Clementina Lang, Alice Panchaud, Christoph R. Meier, Julia Spoendlin

**Affiliations:** 1Hospital Pharmacy, University Hospital Basel, 4031 Basel, Switzerland; caroleanna.marxer@usb.ch (C.A.M.); marlenesusanne.rauch@usb.ch (M.R.); christoph.meier@usb.ch (C.R.M.); 2Basel Pharmacoepidemiology Unit, Division of Clinical Pharmacy and Epidemiology, Department of Pharmaceutical Sciences, University of Basel, 4056 Basel, Switzerland; 3Department of Gynecology, University Hospital Zurich, 8091 Zurich, Switzerland; clementina.lang@usz.ch; 4Institute of Primary Health Care (BIHAM), University of Bern, 3012 Bern, Switzerland; alice.panchaud@chuv.ch; 5Service of Pharmacy, Lausanne University Hospital and University of Lausanne, 1011 Lausanne, Switzerland; 6Materno-Fetal and Obstetrics Research Unit, Department “Femme-Mère-Enfant”, University Hospital, 1011 Lausanne, Switzerland

**Keywords:** pregnancy, chronic diseases, obstetric outcomes, observational research, Hospital Medical Statistics, Medizinische Statistik der Krankenhäuser, Swiss health data

## Abstract

The prevalence of chronic diseases during pregnancy and adverse maternal obstetric outcomes in Switzerland has been insufficiently studied. Data sources, which reliably capture these events, are scarce. We conducted a nationwide observational cross-sectional study (2012–2018) using data from the Swiss Hospital Medical Statistics (MS) dataset. To quantify the recording of chronic diseases and adverse maternal obstetric outcomes during delivery in hospitals or birthing centers (delivery hospitalization), we identified women who delivered a singleton live-born infant. We quantified the prevalence of 23 maternal chronic diseases (ICD-10-GM) and compared results to a nationwide Danish registry study. We further quantified the prevalence of adverse maternal obstetric outcomes (ICD-10-GM/CHOP) during the delivery hospitalization and compared the results to existing literature from Western Europe. We identified 577,220 delivery hospitalizations, of which 4.99% had a record for ≥1 diagnosis of a chronic disease (versus 15.49% in Denmark). Moreover, 13 of 23 chronic diseases seemed to be substantially under-recorded (8 of those were >10-fold more frequent in the Danish study). The prevalence of three of the chronic diseases was similar in the two studies. The prevalence of adverse maternal obstetric outcomes was comparable to other European countries. Our results suggest that chronic diseases are under-recorded during delivery hospitalizations in the MS dataset, which may be due to specific coding guidelines and aspects regarding whether a disease generates billable effort for a hospital. Adverse maternal obstetric outcomes seemed to be more completely captured.

## 1. Introduction

The average maternal age in Switzerland continuously increased between 1970 (27.8 years) and 2020 (32.2 years) [1,2]. This trend has been observed in most high-income countries and is driven by the increasing engagement of women in higher education and career building, improved medical care during pregnancy, and advances in assisted reproductive technology [3]. Older maternal age is associated with a higher prevalence of chronic diseases during pregnancy [4], which imposes potential risks to the mother and to the unborn child [5,6,7,8,9,10,11,12,13]. Most drugs to treat diseases of pregnant women are used off-label and based on weak evidence of safety during pregnancy [14]. On the other hand, untreated chronic diseases can also increase the risk of adverse materno-fetal outcomes [15]. To improve the medical care of women with chronic diseases during pregnancy in Switzerland, it is important to know the proportion of women affected and to determine the risks of adverse obstetric outcomes.

A Danish nationwide registry-based study in 107,870 pregnancies of women who delivered between 2009 and 2013 reported that 15.49% had ≥1 diagnosis of a chronic disease recorded prior to delivery [16].

In Switzerland, the prevalence of chronic diseases in pregnant women has been insufficiently evaluated (lack of studies). Electronic health databases are under-used, although the Swiss Federal Council declared the use of electronic health data for public health research a key priority in 2013 (Health2020 [17], Health2030 [18]). Digitalization in the Swiss health care sector remains under-developed and highly fragmented, and uncertainty around legal aspects often prevents the linking of different data sources, to study medical questions of public health importance [19].

We aimed to conduct a cross-sectional study in the Swiss Hospital Medical Statistics (MS) dataset, to quantify the recording of chronic diseases and adverse maternal obstetric outcomes during delivery in hospitals or birthing centers (delivery hospitalization) between 2012 and 2018. In order to infer on the completeness of the evaluated chronic diseases and adverse maternal obstetric outcomes, we aimed to compare our results to other studies from Western Europe with a similar sociodemographic distribution.

## 2. Methods

### 2.1. Study Design, Data Source

We performed a nationwide cross-sectional study (2012–2018) using anonymized patient-level inpatient data from the Swiss MS dataset, which is provided by the Swiss Federal Statistical Office (FSO) [20]. The dataset includes nationally collected electronic information on all inpatient stays at hospitals, birthing centers, clinics (including psychiatric clinics), and rehabilitation centers in Switzerland since 1998, with the participation of 100% of all hospitals since 2014 [21].

The MS dataset also contains delivery hospitalizations carried out in Switzerland, but of women living abroad, but does not contain deliveries at home, in hospitals, or birthing centers, if they were performed in an outpatient setting [1].

The MS dataset includes information on sex, age (5-year categories), region of residence, length of inpatient stay, other administrative variables (e.g., type of insurance) and up to 50 diagnoses (International Statistical Classification of Diseases 10th Revision German Modification, ICD-10-GM) and 100 procedure codes (Swiss Classification of Operations, CHOP) per stay [21].

The original purpose of collecting MS data was the epidemiological surveillance of the Swiss hospital population, to monitor services, including quality and cost, and to provide an overview of the inpatient care situation by region. Since 2012, MS data have also served as the basis to adjust the billing codes of the bundled case-based reimbursement system in inpatient care (Swiss Diagnosis Related Groups, SwissDRG) on a yearly basis. Generally, all billable health services are captured in the MS dataset [22].

### 2.2. Study Population

Our study population included all hospital stays (including at birthing centers) of women who delivered a singleton live-born infant in an inpatient setting in Switzerland (delivery hospitalization) between 1 January 2012 and 31 December 2018. We extracted all deliveries based on a recorded ICD-10-GM code indicating delivery and excluded deliveries with an ICD-10-GM code indicating stillbirth and/or multiple birth (codes in Appendix A). An individual woman may have contributed more than one delivery hospitalization to the study population. A flow chart of cohort enrolment is shown in Figure 1.

### 2.3. Captured Variables

We captured age at the delivery hospitalization (5-year categories), the length of the delivery hospitalization, and the type of admission (emergency, planned, unknown; Table 1).

### 2.4. Chronic Diseases

We captured 23 chronic diseases (Table 2) recorded as a primary or secondary diagnostic ICD-10-GM codes (Appendix A) during the delivery hospitalization. To increase comparability, we evaluated the same chronic diseases as reported in the previously mentioned national Danish registry-study [16].

### 2.5. Obstetric Outcomes

All codes used to define adverse maternal obstetric outcomes are shown in Appendix A. We captured the mode of delivery, recorded as a primary or secondary procedure (CHOP) code (Table 3). We divided cesarean sections after 2014 into primary (planned) and secondary (emergency) cesarean sections. No specific codes were available before 2014. Length of gestation in completed weeks was captured in the following categories: preterm (<37), term (37–41), and post-term (>41) delivery. Preterm delivery was further divided into <25, 26–33, and 34–36 completed weeks of gestation.

We further captured other adverse maternal obstetric outcomes that have previously been associated with maternal chronic disease, recorded during the delivery hospitalization as a primary or secondary ICD-10-GM code [6,7,8,9,10,11,12,13].

### 2.6. Statistical Analysis

We used descriptive statistics to present demographics and characteristics, as well as the recorded prevalence of chronic diseases and adverse maternal obstetric outcomes overall, by maternal age categories, and by calendar year. A test for normality was done for continuous variables.

In order to infer the completeness of our evaluated chronic diseases and adverse maternal obstetric outcomes, we compared the prevalence of recorded pre-existing chronic diseases in the MS dataset (2012–2018) to the prevalence reported in the national Danish registry study (2009–2013) [16]. We further compared the prevalence of adverse maternal obstetric outcomes to the currently available literature from Western Europe with a similar sociodemographic distribution.

All analyses were conducted using Python 3.0 [23].

## 3. Results 

We extracted 577,220 delivery hospitalizations resulting in a singleton live-born infant (Figure 1). In total, 8.91% of mothers were aged < 25 years, 63.66% between 25 and 34 years, and 27.43% ≥ 35 years. The mean length of stay was 4.33 days (SD = 2.87; Table 1).

### 3.1. Chronic Diseases

The prevalence of recorded diagnoses of maternal chronic diseases during the delivery hospitalization and a comparison with the Danish registry study is shown in Table 2. We observed ≥1 recorded diagnosis of one of the 23 evaluated chronic diseases during 4.99% of delivery hospitalizations. The prevalence increased between 2012 and 2018 (3.30–6.88%), but was lower than in the Danish study (15.49% between 2009 and 2013; Table 2; Appendix A). In total, 13 of 23 chronic diseases were substantially less prevalent in our study population compared to the Danish study (Table 2; 8 of those were >10-fold more frequent in the Danish study. These included polycystic ovary syndrome, anxiety and personality disorders, mood disorders, chronic lung disease (including asthma), rheumatoid arthritis, schizophrenia or other paranoid psychoses, ischemic heart diseases, cerebrovascular diseases, inflammatory bowel disease, hypertension, epilepsy, multiple sclerosis, and diabetes mellitus type 1 or 2. 

The other 10 chronic diseases were similarly prevalent in both datasets, but seven of them (polyglandular dysfunction, parathyroid disorders, Cushing syndrome, polyarthritis nodosa, systemic lupus erythematosus, human immunodeficiency virus (HIV), and atherosclerosis) had a prevalence of ≤0.07% in both datasets. The remaining three chronic diseases were prevalent in >0.07% in both datasets and included thyroid disorders, coagulation disorders, and non-ischemic chronic heart diseases (e.g., cardiomyopathies, heart valve diseases, atrial fibrillation, cardiac arrhythmias).

### 3.2. Obstetric Outcomes

The prevalence of mode of delivery, length of gestation, and recorded adverse maternal obstetric outcomes during the delivery hospitalization is shown in Table 3 (Appendix A: all variables by maternal age category).

**Table 3 ijerph-19-07922-t003:** Prevalence of mode of delivery, length of gestation, and other recorded adverse maternal obstetric outcomes during the delivery hospitalization in the MS dataset. All variables by maternal age category are shown in Appendix A.

	Age	2012–2018	2012	2013	2014	2015	2016	2017	2018
All	All	577,220 (100.00)	79,184 (100.00)	79,945 (100.00)	81,995 (100.00)	83,306 (100.00)	84,619 (100.00)	83,902 (100.00)	84,269 (100.00)
	<25	51,437 (8.91)	8230 (10.39)	8048 (10.07)	7540 (9.20)	7430 (8.92)	7288 (8.61)	6624 (7.90)	6277 (7.45)
	25–34	367,468 (63.66)	50,308 (63.53)	50,800 (63.54)	52,333 (63.83)	53,208 (63.87)	53,910 (63.71)	53,426 (63.68)	53,483 (63.47)
	≥35	158,315 (27.43)	20,646 (26.07)	21,097 (26.39)	22,122 (26.98)	22,668 (27.21)	23,421 (27.68)	23,852 (28.43)	24,509 (29.08)
**Mode of delivery**									
Non-instrumental vaginal delivery ^a^	All	321,572 (55.71) ^a^	39,857 (50.34) ^a^	44,713 (55.93)	45,784 (55.84)	46,961 (56.37)	47,930 (56.64)	47,999 (57.21)	48,328 (57.35)
Instrumental vaginal delivery									
Any instrumental vaginal delivery	All	65,991 (11.43)	9266 (11.70)	9132 (11.42)	9307 (11.35)	9463 (11.36)	9560 (11.30)	9620 (11.47)	9643 (11.44)
Forceps delivery	All	6546 (1.13)	1021 (1.29)	995 (1.25)	944 (1.15)	902 (1.08)	965 (1.14)	817 (0.97)	902 (1.07)
Vacuum delivery	All	59,579 (10.32)	8280 (10.46)	8159 (10.21)	8367 (10.20)	8594 (10.32)	8611 (10.18)	8820 (10.51)	8748 (10.38)
Cesarean section									
Any cesarean section ^b^	All	185,680 (32.17)	25,662 (32.41)	25,977 (32.49)	26,953 (32.87)	27,035 (32.45)	27,319 (32.29)	26,403 (31.47)	26,331 (31.25)
Primary cesarean section	All	NA	NA	NA	7864 (15.03)	7912 (14.87)	7947 (14.74)	7592 (14.21)	7311 (13.67)
Secondary cesarean section	All	NA	NA	NA	12,592 (15.36)	12,762 (15.32)	12,872 (15.21)	12,673 (15.11)	128,61 (15.26)
**Length of gestation ^c^**									
All preterm deliveries (<37 completed weeks)	All	23,724 (4.11)	3607 (4.56)	3497 (4.37)	3367 (4.12)	3335 (4.00)	3326 (3.93)	3271 (3.90)	3321 (3.94)
Preterm delivery <25 completed weeks (extremely preterm)	All	1314 (0.23)	167 (0.21)	195 (0.24)	202 (0.25)	184 (0.22)	181 (0.21)	199 (0.24)	186 (0.22)
Preterm delivery 26–33 completed weeks (very preterm)	All	5943 (1.03)	895 (1.13)	834 (1.04)	851 (1.04)	877 (1.05)	810 (0.96)	799 (0.95)	877 (1.04)
Preterm delivery 34–36 completed weeks (moderate-late preterm)	All	16,478 (2.86)	2548 (3.22)	2469 (3.09)	2315 (2.82)	2275 (2.73)	2336 (2.76)	2275 (2.71)	2260 (2.68)
Term delivery (37–41 completed weeks)	All	493,106 (85.43)	69,111 (87.28)	68,256 (85.38)	69,897 (85.25)	70,769 (84.95)	71,613 (84.63)	71,629 (85.37)	71,831 (85.24)
Postterm delivery (>41 completed weeks)	All	58,856 (10.20)	6328 (7.99)	8085 (10.11)	8668 (10.57)	8901 (10.69)	8855 (10.47)	8954 (10.67)	9065 (10.76)
**Other adverse maternal obstetric outcomes**									
≥1 recorded other adverse maternal obstetric outcome	All	100,501 (17.41)	11,752 (14.84)	12,468 (15.60)	13,219 (16.12)	14,299 (17.16)	15,163 (17.92)	16,281 (19.41)	17,319 (20.55)
Pre-eclampsia/HELLP-syndrome	All	10,355 (1.79)	1183 (1.49)	1444 (1.81)	1426 (1.74)	1457 (1.75)	1571 (1.86)	1628 (1.94)	1646 (1.95)
Placental abruption (abruptio placentae)/ischemic placental disease	All	3227 (0.56)	369 (0.47)	416 (0.52)	446 (0.54)	450 (0.54)	516 (0.61)	516 (0.62)	514 (0.61)
Postpartum hemmorhage	All	45,899 (7.95)	5457 (6.89)	5671 (7.09)	5996 (7.31)	6506 (7.81)	6965 (8.23)	7479 (8.91)	7825 (9.29)
Gestational diabetes	All	37,990 (6.58)	3932 (4.97)	4349 (5.44)	4781 (5.83)	5435 (6.52)	5854 (6.92)	6451 (7.69)	7188 (8.53)
Gestational hypertension	All	5889 (1.02)	922 (1.16)	759 (0.95)	857 (1.05)	831 (1.00)	839 (0.99)	818 (0.98)	863 (1.02)
Coagulopathy	All	1702 (0.30)	106 (0.13)	80 (0.10)	106 (0.13)	136 (0.16)	289 (0.34)	459 (0.55)	526 (0.62)
Sepsis	All	1722 (0.30)	210 (0.27)	202 (0.25)	231 (0.28)	243 (0.29)	258 (0.31)	309 (0.37)	269 (0.32)
Shock	All	1841 (0.32)	229 (0.29)	304 (0.38)	241 (0.29)	218 (0.26)	275 (0.33)	260 (0.31)	314 (0.37)
Status asthmaticus	All	914 (0.16)	119 (0.15)	114 (0.14)	128 (0.16)	150 (0.18)	130 (0.15)	121 (0.14)	152 (0.18)
Status epilepticus	All	1237 (0.21)	155 (0.20)	177 (0.22)	162 (0.20)	186 (0.22)	177 (0.21)	188 (0.22)	192 (0.23)
Acute heart failure	All	276 (0.05)	34 (0.04)	33 (0.04)	34 (0.04)	30 (0.04)	34 (0.04)	52 (0.06)	59 (0.07)
Acute renal failure	All	138 (0.02)	17 (0.02)	14 (0.02)	17 (0.02)	14 (0.02)	8 (0.01)	29 (0.04)	39 (0.05)
Acute liver failure	All	9 (0.00)	NR	NR	NR	NR	NR	NR	NR
Acute myocardial infarction	All	8 (0.00)	NR	NR	NR	NR	NR	NR	NR
Acute respiratory distress syndrome/respiratory failure	All	150 (0.03	10 (0.01)	25 (0.03)	15 (0.02)	24 (0.03)	27 (0.03)	29 (0.04)	20 (0.04)
Coma	All	15 (0.00)	NR	NR	NR	NR	NR	NR	NR
Delirium	All	12 (0.00)	NR	NR	NR	NR	NR	NR	NR
Puerperal cerebrovascular disorders	All	119 (0.02)	18 (0.02)	8 (0.01)	18 (0.02)	12 (0.01)	22 (0.03)	16 (0.02)	25 (0.03)
Pulmonary edema	All	108 (0.02)	22 (0.03)	13 (0.02)	10 (0.01)	20 (0.02)	15 (0.02)	13 (0.02)	15 (0.02)
Pulmonary embolism	All	144 (0.03)	23 (0.03)	23 (0.03)	26 (0.03)	19 (0.02)	17 (0.02)	10 (0.01)	26 (0.03)
Maternal mortality	All	31 (0.01)	NR	5 (0.01)	NR	6 (0.01)	NR	5 (0.01)	9 (0.01)
Stay in ICU (missing: 103)	All	3841 (0.67)	383 (0.48)	915 (1.15)	488 (0.60)	562 (0.68)	494 (0.58)	513 (0.61)	486 (0.58)

Abbreviations: MS = hospital medical statistics; NA = not available (no CHOP codes available in 2012 and 2013); NR = not reported (because cell size <5 patients); HELLP = hemolysis, elevated liver enzymes and low platelets; ICU = intensive care unit; ^a^ Non-instrumental vaginal delivery: The procedure code 74.91 was not mandatory to record in 2012, which affected the numbers in 2012 and 2012–2018; ^b^ Remaining pregnancies are with other or unknown type of cesarean section than primary or secondary cesarean section; ^c^ Remaining pregnancies are with unknown length of gestation (either a ICD-10-GM code O09.9! or no ICD-10-GM which indicates the length of gestation).

The majority of infants were born via non-instrumental vaginal delivery (55.71%), followed by cesarean section (32.17%) and instrumental vaginal delivery (11.43%).

Preterm delivery was recorded in 4.11%, term delivery in 85.43%, and post-term delivery in 10.20% of all delivery hospitalizations (0.26% unknown).

We observed ≥1 recorded diagnosis of another adverse maternal obstetric outcome during 17.41% of delivery hospitalizations with an increasing proportion over time (14.84% in 2012 to 20.55% in 2018). Gestational diabetes was recorded during 6.58%, and gestational hypertension during 1.02% of delivery hospitalizations, both more frequently at older age (≥35 years vs. <25 years: 8.31% vs. 5.03%, 1.23% vs. 0.80%). Pre-eclampsia/HELLP-syndrome was recorded in 1.79% of delivery hospitalizations, with the highest proportion in women ≥35 years (2.00%). Placental abruption was recorded during 0.56% of delivery hospitalizations and increased with age (<25: 0.47%, ≥35: 0.63%). Postpartum hemorrhage was recorded during 7.95% of delivery hospitalizations and was highest in women <25 years (8.21%). Coagulopathy and sepsis were both recorded in 0.30% of delivery hospitalizations. Shock, status asthmaticus, and status epilepticus were recorded in between 0.16% and 0.32%, and other adverse maternal obstetric outcomes in ≤0.05% of delivery hospitalizations. During 0.67% of delivery hospitalizations, the mother was transferred to an intensive care unit (ICU).

## 4. Discussion

This Swiss study used nationwide inpatient data to evaluate the prevalence of the recording of maternal chronic diseases and adverse maternal obstetric outcomes during hospitalization for a delivery. The recording of 13 of 23 evaluated chronic diseases was substantially lower than in the Danish registry study. In our dataset, at least one chronic disease was recorded during 4.99% of delivery hospitalizations, whereas a nationwide Danish registry study reported a prevalence of 15.49%. On the other hand, recording of adverse maternal obstetric outcomes seemed to be more complete than chronic diseases.

Evidence on the prevalence of chronic diseases during pregnancy and of adverse maternal obstetric outcomes in Switzerland is of great public health relevance, especially given the continuously increasing average maternal age [1,2]. Most chronic diseases impose a relevant and often incompletely understood risk to pregnant women and the unborn infant [6,7,8,9,10,11,12,13]. This requires increased monitoring during pregnancy and delivery [24]. The 23 evaluated chronic diseases are of clinical relevance to the treating physicians during the delivery hospitalizations, and typically do get recorded in the electronic medical record. 

However, the substantially lower prevalence of the majority of these diseases in the MS dataset suggests that they typically do not receive sufficient therapeutic or diagnostic attention or increased nursing care to qualify as separate billable records. In our study population, only 4.99% of delivery hospitalizations had a recorded diagnosis for a chronic disease between 2012 and 2018, whereas 15.49% had a chronic disease recorded between 2009 and 2013 in the nationwide Danish study. 

Survey studies from the US and Germany suggested an even higher prevalence of pre-existing chronic diseases (26.6% and 21.4%) in pregnant women, but direct comparison is not possible, due to methodological differences and the different diseases that were evaluated [6,25]. 

Psychiatric diseases, chronic lung diseases (including asthma), rheumatoid arthritis, ischemic heart diseases, and cerebrovascular diseases were recorded more than 10-fold more frequently in the Danish study than in our MS study population. On the other hand, thyroid disorders, coagulation disorders, and non-ischemic chronic heart diseases were recorded similarly frequently in both studies. From a clinical perspective, the latter do not pose a systematically bigger risk to the mother or the unborn child during delivery, suggesting that the recording of chronic diseases in the MS dataset was driven by specificities of the SwissDRG coding system [22]. These findings suggest that MS data are not an ideal data source to study the prevalence of diagnoses during pregnancy, which are not the main cause for hospitalization and/or do not require substantial treatment or nursing care during the respective hospital stay (this probably applies to other target populations, which have not been evaluated). 

The Danish study used national hospital data from the linked Danish National Registry of Patients (DNRP) and from the Danish Psychiatric Central Register (DPCR), which capture diagnoses, treatments, and examinations from hospital encounters. In contrast to our study, they capture diagnoses from inpatient and outpatient hospital care, and diagnoses were extracted longitudinally over a time period of 10 years prior to delivery [16]. Thus, it can be assumed that the reported prevalence of chronic diseases in this Danish study is close to the true prevalence of diagnosed diseases during pregnancy in the Danish population. Such an exact estimation is possible due to the advanced digitalization in the centralized Danish healthcare system. The provision of health care of every Danish resident has been recorded in different linkable health registries over many decades (e.g., DNRP: 1977) [26,27]. In Swiss MS data, it is possible to capture longitudinal inpatient data over time for an individual, but most chronic diseases in young women do not lead to inpatient hospitalizations [28]. A national Swiss data source, which systematically captures diagnoses from outpatient care is desirable, but does not exist. 

Future studies need to evaluate if chronic diseases are more reliably recorded in the electronic medical records (EMR) maintained by individual hospitals. The EMR depict the true situation in clinical care, and are not censored based on whether or not a diagnosis is billable according to SwissDRG coding rules. However, it needs to be considered that the hospitals that provide EMR for research purposes may not be representative of all deliveries in Switzerland.

Adverse maternal obstetric outcomes were more completely captured in the MS dataset, indicating that the MS dataset may be more accurate to study the prevalence of acute conditions during delivery hospitalization. Mode of delivery was recorded in ≥99.80%, and length of gestation in ≥99.70%, of delivery hospitalizations. The lower proportion of preterm delivery (4.11%) compared to the 5.4% reported by the Swiss Vital Statistics (2014) suggests that delivery hospitalization seems unsuitable to determine prematurity [29]. Future studies need to evaluate if the records of the newborns (vs of the mother) in the MS dataset provide more accurate information on prematurity.

The proportion of women with recorded gestational hypertension (1.02%) was similar to the results of a Swedish nationwide registry-based study (2007–2012) including 555,446 deliveries (1.1%) [30].

Gestational diabetes was more frequent (6.58%) in our study than in two prior Swiss studies at the Lausanne University hospital (retrospective study, 2.7% of 5788 women between 2000 and 2002) [31] and the cantonal hospital of Fribourg (prospective screening study, 4.8% of 1042 women between 2004 and 2005) [32]. This may be explained by the introduction of new screening criteria for gestational diabetes by the International Association of Diabetes and Pregnancy Study Groups in 2010 [33]. A retrospective study at the University Hospital Basel (Switzerland) reported a prevalence of gestational diabetes after gestational week 22 of 3.3% in 2008 and 2009, and of 11.3% between 2010 and 2013. However, given the tertiary care setting, this study population was likely more pre-selected than our study population [34].

The proportion of postpartum hemorrhage in the MS dataset was similar to results of a large Danish registry-based study in 403,092 pregnant women between 2004 and 2010 (6.3%, vs. 7.95% in our study).

Placental abruption was recorded in 0.60% of delivery hospitalizations in the MS data, which is slightly higher compared to a review article based on five studies from Scandinavian countries (0.38–0.51%) [35]. Country-specific demographic differences may explain this finding, as the average maternal age in Scandinavia is younger than in Switzerland [35]. However, other unknown factors may play a role as well.

Pre-eclampsia/HELLP-syndrome was recorded during 1.79% of delivery hospitalizations. A prior Swiss study in 1300 pregnant women, who were screened for pre-eclampsia in routine obstetrical care during trimester 2 and 3 (2008–2011), reported a prevalence of 2.31% [36]. On the other hand, two nationwide studies from Denmark [9] and Sweden [30] reported a prevalence of pre-eclampsia of 4.0% (403,092 pregnancies between 2004 and 2010) and 2.9% (555,446 pregnancies between 2007 and 2012). Pre-eclampsia is likely to have been reliably captured in the MS dataset, given that this diagnosis entails substantial extra clinical care, including monitoring in an intermediate or intensive care unitfor several days. The reasons for the different reported prevalence of pre-eclampsia in different countries could be manifold and will have to be followed up in further studies.

When comparing the prevalence of maternal obstetric outcomes between countries and data sources, differences in outcome definitions and changes in diagnostic guidelines over time may play a role [34]. Furthermore, differences in the demographics of the underlying population, as well as in the local health care systems need to be considered. 

In summary, adverse maternal obstetric outcomes were recorded more completely than chronic diseases during delivery hospitalizations. However, the results need to be replicated in other Swiss data sources to confirm that the reported numbers represent the true prevalence of obstetric outcomes in Switzerland. Linkage of MS data to other Swiss data sources, such as the Swiss health care claims data, would allow following pregnant women over time and studying the impact of in utero exposure to drugs and other factors on the risk of obstetric outcomes. Claims data by the Swiss health insurance Helsana have been used to evaluate drug utilization during pregnancy in Switzerland [19,37], but they do not provide granular diagnostic inpatient information. However, concerns over data privacy and the legal vacuum around the handling of electronic health data for academic research currently prevents the linkage of different data sources. An official framework that provides guidance on how to securely and effectively use Swiss health data to the benefit of the Swiss population is needed and desirable. 

Our study has several strengths. First, MS data allowed complete identification of inpatient deliveries in Switzerland during our study period. Second, given that 98.1% of deliveries in Switzerland take place in an inpatient setting, our results are generalizable to the pregnancy population of Switzerland during the study period.

The following limitations need to be considered. First, we were not able to stratify our analyses by parity, because this information is not available in our dataset. Second, further differences between Switzerland and Denmark may limit the comparability of recorded chronic diseases (e.g., possible differences in diagnostic approaches, in the utilization of medical services, as well as in the local health care system; younger maternal age in Denmark [38]). However, correction for such factors may account for a small part of the observed differences in prevalence, but it would most certainly not explain the large difference in the recording of chronic diseases (4.99% vs. 15.49%). Third, Swiss data has limited potential for studying the prevalence of rare maternal obstetric outcomes (e.g., acute myocardial infarction), due to the relatively small population of Switzerland (8.5 mio in 2018). Fourth, the observed increase in the prevalence of recorded diagnoses of chronic diseases (3.30% with any chronic disease in 2012 to 6.88% in 2018) and maternal obstetric outcomes (14.84% in 2012 to 20.55% in 2018) was still ongoing at the end of the study period. Increasing maternal age may play a role [1,2], but the large part of the observed increase was most likely driven by differences in the speed of implementation of the SwissDRG system across institutions and changes in national coding and/or diagnostic guidelines over time. Ongoing changes in data quality over time are a common problem in electronic health databases, given the ongoing advances in digitalization worldwide [16]. 

## 5. Conclusions

In conclusion, this study highlights the need to validate Swiss data sources for specific study questions and study populations. Our results suggest that MS data is not a suitable data source to study chronic diseases in pregnant women, but may capture the prevalence of adverse maternal obstetric outcomes more accurately. However, the results need to be confirmed in other Swiss data sources. A lack of legal regulations currently prevents the linkage of MS data to external Swiss datasets, such as claims data, which would allow studying questions of great public health importance. An official framework to provide guidance on how to securely and effectively use and link different Swiss health data sources, for the benefit of the Swiss population, is needed.

## Figures and Tables

**Figure 1 ijerph-19-07922-f001:**
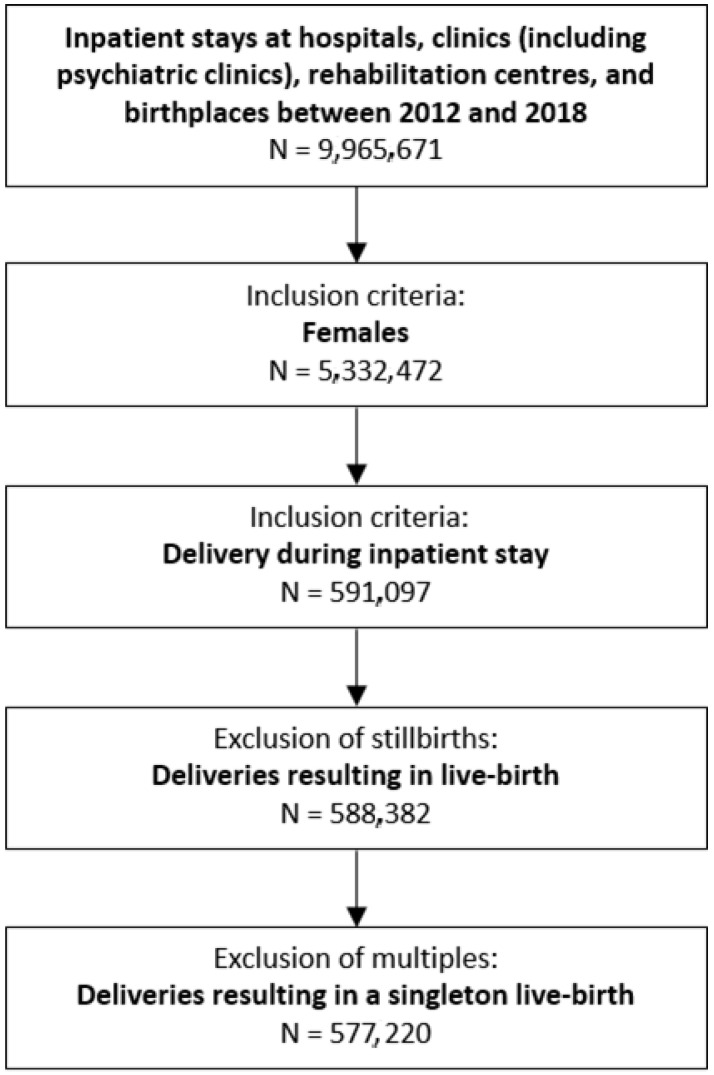
Flow chart of cohort enrolment (numbers separately for each year in Appendix A).

**Table 1 ijerph-19-07922-t001:** Demographics and characteristics of the study population. All variables by maternal age category are shown in Appendix A.

	2012–2018	2012	2013	2014	2015	2016	2017	2018
All	577,220 (100.00)	79,184 (100.00)	79,945 (100.00)	81,995 (100.00)	83,306 (100.00)	84,619 (100.00)	83,902 (100.00)	84,269 (100.00)
Age at delivery hospitalization (years)								
<25	51,437 (8.91)	8230 (10.39)	8048 (10.07)	7540 (9.20)	7430 (8.92)	7288 (8.61)	6624 (7.90)	6277 (7.45)
25–34	367,468 (63.66)	50,308 (63.53)	50,800 (63.54)	52,333 (63.83)	53,208 (63.87)	53,910 (63.71)	53,426 (63.68)	53,483 (63.47)
≥35	158,315 (27.43)	20,646 (26.07)	21,097 (26.39)	22,122 (26.98)	22,668 (27.21)	23,421 (27.68)	23,852 (28.43)	24,509 (29.08)
Length of delivery hospitalization (days), mean (SD) (missing: 31)	4.33 (2.87)	4.567 (2.89)	4.467 (2.64)	4.392 (3.41)	4.329 (2.81)	4.261 (2.74)	4.179 (2.77)	4.134 (2.75)
Type of hospital admission								
Emergency	255,927 (44.34)	32,090 (40.53)	32,828 (41.06)	35,526 (43.33)	38,658 (46.41)	38,562 (45.57)	38,571 (45.97)	39,692 (47.10)
Planned	317,449 (55.00)	46,678 (58.95)	46,620 (58.32)	45,942 (56.03)	44,138 (52.98)	45,443 (53.73)	44,741 (53.33)	43,887 (52.08)
Unknown	3844 (0.67)	416 (0.53)	497 (0.62)	527 (0.64)	510 (0.61)	614 (0.73)	590 (0.70)	690 (0.82)

Abbreviations: SD = standard deviation.

**Table 2 ijerph-19-07922-t002:** Prevalence of recorded chronic diseases in the MS dataset compared to the Danish study sorted by descending relative difference (relative difference = prevalence_Danish study_/prevalence_MS dataset_).

	Prevalence in the MS Dataset (2012–2018, Parity = Unknown) *n*= 577,220		Prevalence in the Danish Study (2009–2013, Parity = 1) *n* = 107,870	
	*n*	%	*n*	%
≥1 recorded chronic disease	28,795	4.99	16,709	15.49
Polycystic ovary syndrome	71	0.01	1835	1.70
Anxiety and personality disorders	507	0.09	3076	2.85
Mood disorders	594	0.10	2305	2.14
Chronic lung disease, including asthma	937	0.16	3443	3.19
Rheumatoid arthritis	226	0.04	786	0.73
Schizophrenia/other paranoid psychoses	259	0.04	592	0.55
Ischemic heart diseases	44	0.01	124	0.11
Cerebrovascular diseases	119	0.02	240	0.22
Inflammatory bowel disease	645	0.11	1177	1.09
Hypertension	499	0.09	740	0.69
Epilepsy	1237	0.21	1056	0.98
Multiple sclerosis	376	0.07	277	0.26
Diabetes mellitus (type 1 or 2)	1915	0.33	692	0.64
Thyroid disorders	15,308	2.65	2471	2.29
Non-ischemic chronic heart diseases	1003	0.17	148	0.14
Coagulation disorders	6169	1.07	793	0.73
Cushing syndrome	63	0.01	41	0.04
HIV	49	0.01	40	0.04
Systemic lupus erythematosus	221	0.04	78	0.07
Polyglandular dysfunction	12	2.08 × 10^−3^	1	0.00
Parathyroid disorders	18	3.12 × 10^−3^	37	0.03
Polyarthritis nodosa	5	8.66 × 10^−4^	9	0.01
Atherosclerosis	6	1.04 × 10^−3^	30	0.03

Abbreviations: *n* = number; MS = Hospital Medical Statistics; HIV = human immunodeficiency virus.

## Data Availability

Original data for this study cannot be provided for legal reasons. Original data can be ordered via the Swiss Federal Statistical Office at https://www.bfs.admin.ch/bfs/de/home/statistiken/gesundheit/erhebungen/ms.html#1042039266 (accessed 27 June 2022).

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
