# Peer review of "Recording of Chronic Diseases and Adverse Obstetric Outcomes during Hospitalizations for a Delivery in the National Swiss Hospital Medical Statistics Dataset between 2012 and 2018: An Observational Cross-Sectional Study"

_ijerph, 2022, doi:10.3390/ijerph19137922_

Round 1
Reviewer 1 Report
The authors made a great work revising the manuscript. It become more clear and aim of the research become feasible.
Only comment now to revise abstract: it should be splitted into the sections.
Author Response
Only comment now to revise abstract: it should be splitted into the sections. - We did not split the abstract into sections, as the author guidelines explicitely ask for an abstract in one paragraph. However, if the editor wishes this to be changed, we are more than happy to change it.Reviewer 2 Report
Presented by the authors manuscript is worth to public, since sucject is extremely important. Please, delete dots before [. The citation should be like this: nbkubiv [nn]. Not: vchtdkhc. [ww].
Author Response
Please, delete dots before [. The citation should be like this: nbkubiv [nn]. Not: vchtdkhc. [ww]. - We went through the document and made sure all references are cited correctly.This manuscript is a resubmission of an earlier submission. The following is a list of the peer review reports and author responses from that submission.
Round 1
Reviewer 1 Report
The reviewed manuscript is a piece of research aimed to evaluate prevalence of chronic diseases and adverse maternal obstetric 64
outcomes using routinely collected data. The data presented at the paper seems to be interesting insight of increasing maternal age and improvements in diagnostic tools but methods used by authors are questionable
Major comments:
1.The main issue for me as a reviewer is the aim of the paper. It is stated, that authors "completeness of recorded diagnoses", but in reality we can see just description of recoded morbidity with comparison of findings with Danish registry. This idea seems strange for me since we cannot draw a conclusion about completeness of Swiss data by comparing them with Danish one without adjustment on all possible confounders. Since authors have not presented information about inter populational diversity, differences in diagnostic approaches, utilization of medical services in the countries, difference in reported prevalence can be explained not only by problems with registration.
2. Authors have not taken into account some other important issues: coding differences between countries, antenatal care organization, organizational issues between registries, proportion of missing data etc. I suggest reformulating the aim and making it in the line with results and conclusions or adjust your analysis including the factors mentioned above.
3. More information on dataset used is needed in Methods
4. No adjusted on parity in comparison between Danish and Swiss data. Authors used in table 2 data from Danish registry that were calculated for 42 358 childbirths with parity = 1 in Denmark, but there is no any limitation for parity in Swiss dataset.
Minor comments:
Many repetition of information between tables and text, for example, lines 136-148 repeat information at table 1, should be revised and shorten.
Table 3 is rather big and should be spitted into 2-3 tables. May be you need to collapse all age categories into one and present detail information at Supplementary materials.
Reviewer 2 Report
In this cross-sectional investigation of singleton live-birth deliveries in Switzerland, approximately 5% of mothers had a record of one or more chronic disease(s). The authors provide comparisons with that from the Danish National Registry of Patients. While prevalence estimation is valuable, there are multiple aspects requiring revision before further consideration for publication.
- There is a mismatch between the primary objective and the approach & results. Authors state the primary objective as evaluating the completeness of recorded diagnosis, however, there are no validation datasets (e.g., records including both inpatient and outpatient care) included in this paper and therefore such aim cannot be evaluated with this dataset. Rather, results are focused on estimating chronic disease prevalence. Please re-phrase the aim or include an appropriate validation dataset.
- The conclusion that chronic diseases are under-reported in inpatient visits cannot be drawn from the results, since prevalence was estimated in one dataset and does not include a validation dataset. The conclusion seems to be based on a comparison with a Danish study, however, results are not directly comparable as the Danish study estimated the prevalence of pregnancy complications using inpatient as well as outpatient visits while the current study is limited to inpatient visits. Also, the study period is different for the two studies and it is unclear if there are any other differences, for example, whether the Danish study also restricted to liveborn singleton pregnancies or had a similar age range. A more appropriate language may be that prevalence of certain chronic conditions were "lower" than that reported in a Danish study, not "under-reported"
- Replacing 0.00 with numbers containing at least one significant figure of a number will be more informative. e.g., 1.72*10-3
- It will be helpful for readers if the authors included references for claims/assertions that are not directly supported by the results. Below are some of many that should include citations
- lines 44-48: Older maternal age is associated with ... increase the risk of adverse materno-fetal outcomes as well.
- line 55: does this mean lack of previous studies or limitations in previous studies? If latter, what are the limitations?
- lines 221-222:From a clinical perspective, the latter do not pose a systematically bigger risk to the mother or the unborn child during delivery
- lines 239-240: In Swiss MS data, it is possible to capture longitudinal data over time for an individual, but most chronic diseases in young women do not lead to inpatient hospitalizations.